# Health worker compliance with severe malaria treatment guidelines in the context of implementing pre-referral rectal artesunate in the Democratic Republic of the Congo, Nigeria, and Uganda: An operational study

Aita Signorell[1,2]*, Phyllis Awor[3], Jean Okitawutshu[1,2,4], Antoinette Tshefu[4], Elizabeth Omoluabi[5], Manuel W. Hetzel[1,2], Proscovia Athieno[3], Joseph Kimera[3], Gloria Tumukunde[3], Irene Angiro[3], Jean-Claude Kalenga[4], Babatunde K. Akano[5], Kazeem Ayodeji[5], Charles Okon[5], Ocheche Yusuf[5], Giulia Delvento[1,2], Tristan T. Lee[1,2], Nina C. Brunner[1,2], Mark J. Lambiris[1,2], James Okuma[1,2], Nadja Cereghetti[1,2], Valentina Buj[6], Theodoor Visser[7], Harriet G. Napier[7], Christian Lengeler[1,2☯], Christian Burri[1,2☯]

1 Swiss Tropical and Public Health Institute, Allschwil, Switzerland, 2 University of Basel, Basel, Switzerland, 3 Makerere University School of Public Health, Kampala, Uganda, 4 Kinshasa School of Public Health, Kinshasa, Democratic Republic of the Congo, 5 Akena Associates, Abuja, Nigeria, 6 UNICEF, New York, New York, United States of America, 7 Clinton Health Access Initiative, Boston, Massachusetts, United States of America

☯ These authors contributed equally to this work.
* aita.signorell@swisstph.ch

## Abstract

### Background

For a full treatment course of severe malaria, community-administered pre-referral rectal artesunate (RAS) should be completed by post-referral treatment consisting of an injectable antimalarial and oral artemisinin-based combination therapy (ACT). This study aimed to assess compliance with this treatment recommendation in children under 5 years.

### Methods and findings

This observational study accompanied the implementation of RAS in the Democratic Republic of the Congo (DRC), Nigeria, and Uganda between 2018 and 2020. Antimalarial treatment was assessed during admission in included referral health facilities (RHFs) in children under 5 with a diagnosis of severe malaria. Children were either referred from a community-based provider or directly attending the RHF.

RHF data of 7,983 children was analysed for appropriateness of antimalarials; a subsample of 3,449 children was assessed additionally for dosage and method of ACT provision (treatment compliance). A parenteral antimalarial and an ACT were administered to 2.7% (28/1,051) of admitted children in Nigeria, 44.5% (1,211/2,724) in Uganda, and 50.3% (2,117/4,208) in DRC. Children receiving RAS from a community-based provider were more

**Data Availability Statement:** All relevant datasets are publicly available on https://zenodo.org/record/7386745.

**Funding:** This study was funded by Unitaid (https://unitaid.org; grant number XM-DAC-30010-CHAIRAS, to all authors). The funders had no role in study design, data collection and analysis, decision to publish, or preparation of the manuscript.

**Competing interests:** The authors have declared that no competing interests exist.

**Abbreviations:** ACT, artemisinin-based combination therapy; aOR, adjusted odds ratio; CARAMAL, Community Access to Rectal Artesunate for Malaria; CHW, community health worker; DRC, Democratic Republic of the Congo; LGA, local government area; PHC, primary health centre; PSS, patient surveillance system; RAS, rectal artesunate; RHF, referral health facility; WHO, World Health Organization.

likely to be administered post-referral medication according to the guidelines in DRC (adjusted odds ratio (aOR) = 2.13, 95% CI 1.55 to 2.92, $P < 0.001$), but less likely in Uganda (aOR = 0.37, 95% CI 0.14 to 0.96, $P = 0.04$) adjusting for patient, provider, caregiver, and other contextual factors. While in DRC, inpatient ACT administration was common, ACTs were often prescribed at discharge in Nigeria (54.4%, 229/421) and Uganda (53.0%, 715/1,349). Study limitations include the unfeasibility to independently confirm the diagnosis of severe malaria due to the observational nature of the study.

## Conclusions

Directly observed treatment was often incomplete, bearing a high risk for partial parasite clearance and disease recrudescence. Parenteral artesunate not followed up with oral ACT constitutes an artemisinin monotherapy and may favour the selection of resistant parasites. In connection with the finding that pre-referral RAS had no beneficial effect on child survival in the 3 study countries, concerns about an effective continuum of care for children with severe malaria seem justified. Stricter compliance with the WHO severe malaria treatment guidelines is critical to effectively manage this disease and further reduce child mortality.

## Trial registration

ClinicalTrials.gov (NCT03568344).

## Author summary

### Why was this study done?

➢ Prompt initiation of parenteral treatment for severe malaria is critical to avoid morbidity and death.

➢ In places where injectable treatment is not available, the World Health Organization (WHO) malaria guidelines recommend that children beyond 6 years of age be administered a single dose of rectal artesunate (RAS) before being referred to a higher-level health facility for parenteral treatment completed with a full course of artemisinin-based combination therapy (ACT).

➢ Little is known about the impact of RAS on severe malaria case fatality in real-world settings as well as about unforeseen and unintended consequences of rolling out this commodity.

### What did the researchers do and find?

➢ We collected data about post-referral antimalarial treatment given to 7,983 children under 5 years of age admitted to referral health facilities (RHFs) in the Democratic Republic of the Congo (DRC), Nigeria, and Uganda and studied compliance with the recommendations issued by the WHO.

➢ We found that a high percentage of children were administered the recommended number of doses of parenteral antimalarial treatment but treatment was rarely to never completed with a full course of ACT, depending on the study country.

➢ Rather than providing ACTs during admission, many health workers discharged children with a prescription for ACTs.

## What do these findings mean?

➢ To avoid incomplete cure, morbidity, and death, it is critical that patients are administered a full course of recommended parenteral treatment that is followed-on by ACT.

➢ Furthermore, administration of rectal and/or parenteral artesunate only constitutes an artemisinin monotherapy and may favour the development or selection of artemisinin resistance.

➢ Unless post-referral treatment is improved, pre-referral RAS is unlikely to have an impact on malaria case fatality.

## Introduction

Malaria deaths result from progression of uncomplicated to severe disease [1]. The risk of dying is highest within the first 24 h after onset of severe symptoms [2]; therefore, prompt initiation of treatment is vital to avert severe morbidity and death. The World Health Organization (WHO) recommends treatment for severe malaria consists of an injectable antimalarial (artesunate, artemether, or quinine) followed by a full course of oral artemisinin-based combination therapy (ACT) [3].

Despite these effective and safe treatment options, many children still die from severe malaria. Two main reasons may be responsible for a fatal outcome: firstly, in several endemic countries, many children never or only belatedly reach the formal health system [4–6]. Secondly, the quality of care that a severely ill child receives is often poor [7–10].

To increase prompt access to essential treatments, the WHO malaria treatment guidelines [3] advise that in situations where parenteral treatment cannot be administered, a single dose of rectal artesunate (RAS) should be given and the child be referred to a health facility where injectable treatment is available. After the WHO prequalification of 2 RAS products in 2018 [11], endemic countries started to scale up RAS distribution [12]. However, evidence is scarce regarding the operational feasibility of incorporating RAS into the continuum of care for severe malaria and the intervention's unanticipated consequences on the overall disease management. In addition, it is unclear how much impact the introduction of RAS will have under real-world circumstances [13].

The Community Access to Rectal Artesunate for Malaria (CARAMAL) project was designed as a large-scale operational study to address these questions [14]. The study aimed to assess healthcare seeking patterns [15], RAS use and acceptance [16], antimalarial treatment received at the various points of contact with the health system, health outcome at day 28 [17], as well as health system costs associated with the roll-out of pre-referral RAS [18]. Contrary to expectations, we found that RAS did not have a beneficial effect on child survival: In the Democratic Republic of the Congo (DRC) and Nigeria, children receiving RAS were more likely to die than those not receiving RAS (adjusted odds ratio (aOR) = 3.06, 95% CI 1.35 to 6.92 and aOR = 2.16, 95% CI 1.11 to 4.21, respectively). Only in Uganda, RAS users were less likely to be dead or sick at follow-up (aOR = 0.60, 95% CI 0.45 to 0.79) [17]. One reason for this finding

may be lower referral completion in children who were administered pre-referral RAS which we found to be the case in all countries in the post-implementation phase [15].

RAS on its own is insufficient to cure severe malaria. It is therefore important to understand the post-referral care and treatment patients receive in referral health facilities (RHFs).

This paper describes severe malaria treatment provided to children below 5 years in RHFs in the context of RAS roll-out and provides evidence for necessary improvements in severe malaria case management in addition to delivering this gap-filling commodity.

## Methods

### Study design and participants

The present results were obtained in the frame of the CARAMAL project, a multicountry observational study on the implementation of quality assured pre-referral RAS by community health workers (CHWs) and primary health centres (PHCs). The details of the design and methods of the CARAMAL project have been published elsewhere [14]. In short, CARAMAL was designed as a pre-post intervention study that started in April 2018. The post-RAS introduction period ran from April/May 2019 until August 2020. The study areas included 3 health zones in the DRC (Ipamu, Kenge, and Kingandu), 3 local government areas (LGAs) of Adamawa State in Nigeria (Fufore, Mayo-Belwa, and Song), and 3 districts in Uganda (Kole, Kwania, and Oyam). Local health authorities with support from UNICEF were responsible for training of healthcare providers, behaviour change and communication activities, and continuous supply of RAS.

The main data collection component of the CARAMAL study was a patient surveillance system (PSS) in which children with suspected severe malaria were provisionally enrolled upon their first contact with CHWs or PHCs (S1 File). Inclusion criteria were aligned with the criteria for administering RAS according to the country guidelines and included age under 5 years, history of fever, plus at least 1 danger sign defining a severe febrile illness episode according to the national iCCM guidelines (not able to drink or feed anything, unusually sleepy or unconscious, convulsions, vomits everything). Following provisional enrolment of an eligible patient, basic information on inclusion criteria, RAS administration and referral was transmitted to the study team by the healthcare worker according to country-specific notification routes, and captured in electronic study forms and registers. Patients who were successfully referred from a CHW or PHC to the RHFs in the study areas were identified and monitored by trained study nurses based at each of the study areas' 25 RHFs. For a comprehensive assessment of severe malaria treatment at included RHFs, we also enrolled children below the age of 5 directly attending such RHFs and diagnosed with severe malaria. Only patients diagnosed with severe malaria by RHF clinicians were included in this analysis; the diagnosis was, however, not verified for correctness. Children receiving outpatient antimalarial treatment at RHFs (mainly Uganda) were not included in this study.

All monitored RHFs were public or private not-for-profit institutions, including health centre level IV and hospitals in Uganda, cottage hospitals in Nigeria, and referral health centres and general reference hospitals in DRC.

A follow-up visit at patients' homes was scheduled 28 days after enrolment for all children enrolled into the study, which included a structured interview about the patient's health status, signs and symptoms of the disease, and treatment seeking.

## Study procedures

Case management information was extracted from patients' hospital records by trained study staff and complemented by direct observation and information obtained from resident hospital staff. Inpatient treatment data was transcribed in real-time on paper forms and then copied onto tablets using structured electronic forms in ODK Collect (https://opendatakit.org/). An updated data collection form implemented 2 (Nigeria) to 4 (DRC, Uganda) months after the roll-out of RAS also included drug type, route, and dates and times of antimalarial therapy, and details of ACT prescription/dispensing at discharge. Health status, as well as the payment scheme for hospitalisation and medication (artesunate and ACT were supposed to be free of charge) were recorded upon hospital release. Information on the use of pre-referral RAS was consolidated from different data sources (enrolment register, RHF and day 28 data collection forms) through the different points of contact with the healthcare system (CHW, PHC, RHF) and from a caregiver's interview on day 28.

## Definitions

The definitions used for measurement of compliance outcomes are shown in Fig 1. Medication appropriateness was defined as administration of recommended parenteral antimalarials, including artemisinin, artemether, or quinine, plus oral ACTs (at least 1 dose of each); medication appropriateness was computed for the entire study population. Treatment compliance was defined as provision of at least 3 doses of an appropriate parenteral antimalarial (artesunate, artemether, or quinine) followed by administration, dispensing, or prescription of an ACT; treatment compliance was assessed for data from a post-implementation subsample for which the updated, more comprehensive data collection form was used.

## Statistical analysis

Analyses were planned in January 2021 after completion of data collection. Results were stratified by country and enrolment location and calculated as overall proportions. Proportions were compared by $\chi^2$ test.

Age, sex, weight, pre-referral RAS administration, treatment seeking, and malaria test result were considered as potential patient-level predictors of medication appropriateness. Caregivers were asked a yes/no question about whether they had to pay any fees either for hospitalisation or for medication; this was analysed as provider-level predictor; contextual predictors comprised study country, study area, and seasonality. Upon peer review of the manuscript, caregiver age was added as an important predictor; since the variable "highest level of education completed" had 51% missing data, it was not included in the model. Potential predictors for medication appropriateness were determined by a logistic regression model with a binary outcome variable equal to 1 for appropriate medication. We report a multivariable model adjusting for all other predictors. All models were based on complete cases. In addition, we also did a sensitivity analysis using multiple imputation methods for weight, which had 12.4% missing values of the total sample.

All analyses were performed using Stata/MP 16.1 (StataCorp, College Station, Texas, United States of America). This study is reported as per the Strengthening the Reporting of Observational Studies in Epidemiology (STROBE) guideline (S2 File).

## Ethics statement

The CARAMAL study protocol was approved by the Research Ethics Review Committee of the World Health Organization (WHO ERC, No. ERC.0003008), the Ethics Committee of the

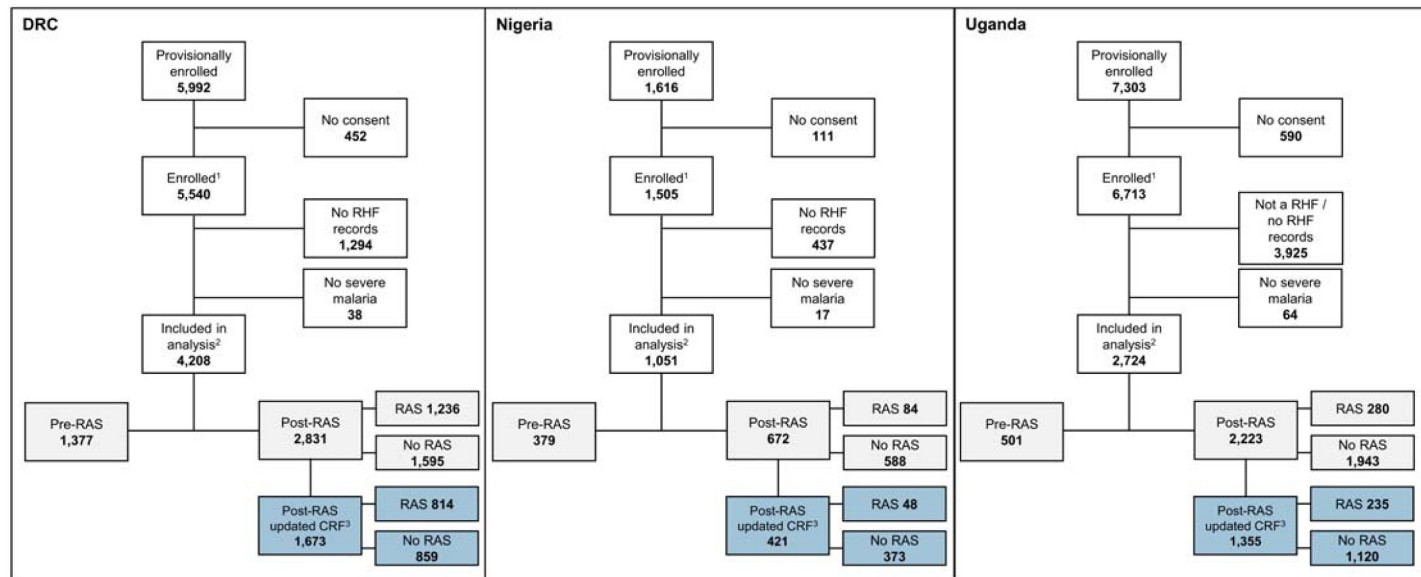

| Definition | | Study period |
|---|---|---|
| Treatment compliance | Medication appropriateness: At least one dose of an injectable antimalarial (artesunate, artemether or quinine) and at least one dose of an ACT (artemether-lumefantrine, artesunate-amodiaquine) while the patient was still hospitalised | Full |
| | Treatment compliance: At least three doses of an injectable antimalarial (artesunate, artemether or quinine), followed by at least one dose of an ACT (artemether-lumefantrine, artesunate-amodiaquine) administered by health workers during admission, or prescribed / dispensed at RHF discharge | Post-implementation (updated data collection tool) |

**Fig 1. Details of analysis dataset and definitions of antimalarial treatment compliance.** Of the 14,911 children enrolled in the CARAMAL study (either by a community-based provider or at the RHF), 7,983 underwent treatment at a RHF and were included in the analysis (2,257 enrolled during the RAS pre-implementation period (pre-RAS), 5,726 enrolled after RAS was rolled out (post-RAS)). A total of 1,600 received pre-referral RAS during the post-RAS phase, 4,126 did not. Medication appropriateness was assessed for the full dataset (highlighted in light grey), treatment compliance was analysed for the post-implementation subsample (shown with blue background). ACT, artemisinin-based combination therapy; CRF, case report form; DRC, Democratic Republic of the Congo; RAS, rectal artesunate; RHF, referral health facility. [1] Enrolled in CARAMAL study by community-based provider or directly enrolled at RHF. [2] Followed up after admission to RHF. [3] Post-RAS subsample with more detailed treatment data.

University of Kinshasa School of Public Health (No. 012/2018), the Health Research Ethics Committee of the Adamawa State Ministry of Health (S/MoH/1131/I), the National Health Research Ethics Committee of Nigeria (NHREC/01/01/2007-05/05/2018), the Higher Degrees, Research and Ethics Committee of the Makerere University School of Public Health (No. 548), the Uganda National Council for Science and Technology (UNCST, No. SS 4534), and the Scientific and Ethical Review Committee of CHAI (No. 112, 21 Nov 2017). The study is registered on ClinicalTrials.gov (NCT03568344). Only patients whose caregivers provided written informed consent were enrolled in the study.

### Inclusivity in global research

Additional information regarding the ethical, cultural, and scientific considerations specific to inclusivity in global research is included in the Supporting information (S3 File).

## Results

### Characteristics of patients

Between April 2018 and July 2020, 14,911 children were provisionally enrolled into the PSS. For 6,928 children caregivers either did not provide consent, children were not admitted to one of the RHFs monitored by the study team, or did not have a diagnosis of severe malaria at admission. Hence, 7,983 children were included in this analysis (Fig 1). For 3,449 (43.2%) of them, more detailed case management data was recorded with an updated data collection form.

Table 1 (full sample) and S1 Table (subsample) show baseline characteristics of the children included in this analysis. A total of 2,381 (29.8%) children undergoing treatment at an RHF were enrolled by a community-based healthcare provider (CHW, PHC) while 5,602 (70.2%) were enrolled directly at an RHF. Among the children enrolled by a community-based provider during the post-RAS implementation period, the proportion who received a dose of pre-referral RAS was higher in DRC (82.6%, 1.188/1,939) and Uganda (73.2%, 123/225) than in Nigeria (44.2%, 80/217).

### Parenteral treatment

Across the full study period, most of the children were treated with an injectable antimalarial at RHFs (DRC 83.7% (3,521/4,208), Nigeria 93.6% (984/1,051), and Uganda 94.8% (2,583/2,724); Table 2). In 86.8% of these cases (6,153/7,088), injectable artesunate was administered. During the post-implementation period, administration of parenteral antimalarials was higher than in the pre-implementation period (Fig 2 and Table 2). In DRC, the use of intravenous quinine was still common (18.3% among all children (N = 4,208)) though it was gradually replaced by artesunate throughout study duration (34.6% (476/1,377) pre-implementation, 10.3% (292/2,831) post-implementation). Strikingly, during both study periods, the use of quinine was relatively more common in children directly attending an RHF compared to community referrals (56.2% (360/641) versus 37.3% (116/311) pre-implementation, 14.1% (179/1,267) versus 8.7% (113/1,302) post-implementation, both $P < 0.001$).

### ACT follow-on treatment

While only 2.7% (28/1,051) of the children in Nigeria received appropriate antimalarials during admission, i.e., an injectable and an ACT, 44.5% (1,211/2,724) did so in Uganda and 50.3% (2,117/4,208) in DRC (Table 2).

The pooled proportion of children receiving appropriate treatment increased significantly between the pre-implementation (21.9%, 494/2,257) and the post-implementation period (50.0%, 2,862/5,726, $P < 0.001$; Fig 2), mainly attributed to an increase of ACT administration among children treated with an injectable antimalarial observed in DRC (46.9 percentage points (258/1,377 pre- versus 1,859/2,831 post-implementation); $P < 0.001$) and Nigeria (3.8 percentage points (1/379 pre- versus 27/672 post-implementation); $P = 0.001$). Meanwhile, there was no difference of follow-on ACT use in Uganda.

### Predictors of appropriate antimalarial medication

While there was no difference in the odds of receiving appropriate antimalarials between community referrals and direct RHF attendances in Uganda and DRC (Table 3), community referrals in DRC were more likely to receive an injectable antimalarial followed by an ACT if they had received RAS pre-referral treatment in the post-implementation period (aOR = 2.13, 95% CI 1.55 to 2.92). In contrast, children in Uganda were less likely to be receiving an appropriate

**Table 1. Summary characteristics of surveyed patients and exposure variables.**

| | DRC | | Nigeria | | Uganda | |
|---|---|---|---|---|---|---|
| | Community enrolments | RHF enrolments | Community enrolments | RHF enrolments | Community enrolments | RHF enrolments |
| | *N* = 1,939 | *N* = 2,269 | *N* = 217 | *N* = 834 | *N* = 225 | *N* = 2,499 |
| | *n* (%) | *n* (%) | *n* (%) | *n* (%) | *n* (%) | *n* (%) |
| **Age (years)** | | | | | | |
| <1 | 428 (22.1) | 475 (20.9) | 19 (8.8) | 91 (10.9) | 46 (20.4) | 495 (19.8) |
| 1–2 | 996 (51.4) | 1,115 (49.1) | 134 (61.8) | 474 (56.8) | 136 (60.4) | 1,377 (55.1) |
| 3 - < 5 | 515 (26.6) | 679 (29.9) | 64 (29.5) | 269 (32.3) | 43 (19.1) | 627 (25.1) |
| **Sex** | | | | | | |
| Female | 931 (48.0) | 1,053 (46.4) | 76 (35.0) | 367 (44.0) | 106 (47.1) | 1,122 (44.9) |
| **RAS implementation period and pre-referral RAS use** | | | | | | |
| Pre-implementation | 501 (25.8) | 876 (38.6) | 36 (16.6) | 343 (41.1) | 57 (25.3) | 444 (17.8) |
| RAS use: yes | 3 (0.6) | 2 (0.2) | 0 (0.0) | 0 (0.0) | 2 (3.5) | 6 (1.4) |
| RAS use: no | 498 (99.4) | 874 (99.8) | 36 (100.0) | 343 (100.0) | 55 (96.5) | 438 (98.7) |
| Post-implementation | 1,438 (74.2) | 1,393 (61.4) | 181 (83.4) | 491 (58.9) | 168 (74.7) | 2,055 (82.2) |
| RAS use: yes | 1,188 (82.6) | 48 (3.5) | 80 (44.2) | 4 (0.8) | 123 (73.2) | 157 (7.6) |
| RAS use: no | 250 (17.4) | 1,345 (96.6) | 101 (55.8) | 487 (99.2) | 45 (26.8) | 1,898 (92.4) |
| **Malaria test**[*] | | | | | | |
| Positive (mRDT or blood slide) | 1,695 (87.4) | 2,007 (88.5) | 197 (90.8) | 740 (88.7) | 222 (98.7) | 2,477 (99.1) |
| Negative/not done | 244 (12.6) | 262 (11.6) | 20 (9.2) | 94 (11.3) | 3 (1.3) | 22 (0.9) |
| **Rainy season**[°] | 1,096 (56.5) | 1,113 (49.1) | 159 (73.3) | 547 (65.6) | 156 (69.3) | 1,723 (69.0) |
| **Drugs payable** | 863 (46.5) | 1,005 (54.3) | 144 (72.0) | 543 (78.1) | 35 (15.6) | 419 (16.8) |
| Missing | 84 (4.3) | 418 (18.4) | 17 (7.8) | 139 (16.7) | 0 (0.0) | 0 (0.0) |
| **Hospitalisation payable** | 874 (47.1) | 859 (46.4) | 33 (16.5) | 76 (10.9) | 10 (4.4) | 394 (15.8) |
| Missing | 84 (4.3) | 418 (18.4) | 17 (7.8) | 139 (16.7) | 0 (0.0) | 0 (0.0) |
| **Age caregiver (years)** | | | | | | |
| <30 | 576 (29.7) | 806 (35.6) | 108 (49.8) | 378 (45.3) | 148 (65.8) | 1,529 (61.2) |
| ≥30 | 1,363 (70.3) | 1,461 (64.5) | 109 (50.2) | 456 (54.7) | 77 (34.2) | 969 (38.8) |
| missing | 0 (0.0) | 2 (0.1) | 0 (0.0) | 0 (0.0) | 0 (0.0) | 1 (0.0) |
| **Education caregiver** | | | | | | |
| Completed secondary education | 563 (59.8) | 560 (63.4) | 36 (33.6) | 95 (29.6) | 12 (8.5) | 177 (11.8) |
| Completed primary education | 183 (19.5) | 186 (21.0) | 18 (16.8) | 76 (23.7) | 64 (45.1) | 575 (38.2) |
| No education | 195 (20.7) | 138 (15.6) | 53 (49.5) | 150 (46.7) | 66 (46.5) | 755 (50.1) |
| Missing | 998 (51.5) | 1,358 (61.0) | 110 (50.7) | 513 (61.5) | 83 (36.9) | 992 (39.7) |
| **Health zone/LGA/District**[**] | | | | | | |
| Kenge DRC/Fufore NG/Kole UG | 617 (31.8) | 883 (38.9) | 71 (32.7) | 309 (37.1) | 74 (32.9) | 618 (24.7) |
| Kingandu DRC/Mayo Belwa NG/Oyam UG | 503 (25.9) | 278 (12.3) | 123 (56.7) | 227 (27.2) | 119 (52.9) | 1,399 (56.0) |
| Ipamu DRC/Song NG/Kwania UG | 819 (42.2) | 1,108 (48.8) | 23 (10.6) | 298 (35.7) | 32 (14.2) | 482 (19.3) |

Number and column % of those with non-missing data, by country and by enrolment location; missing data rows are number and column %.

[*] Severe malaria diagnosis was based on clinical assessment, diagnostic test result may or may not have been considered for the diagnosis.

[°] At time of admission; DRC: October–April; Nigeria: May–October; Uganda: April–October.

[**] Health zones in DRC (Kenge, Kingandu, Ipamu)/LGA in Nigeria (Fufore, Mayo Belwa, Song)/District in Uganda (Kole, Oyam, Kwania).

DRC, Democratic Republic of the Congo; LGA, local government area; mRDT, rapid diagnostic test for malaria; NG, Nigeria; RAS, rectal artesunate; RHF, referral health facility; UG, Uganda.

**Table 2. Administration of antimalarial treatment for severe malaria, by country and enrolment level.**

| | DRC | | | Nigeria | | | Uganda | | |
|---|---|---|---|---|---|---|---|---|---|
| | Community enrolments | RHF enrolments | P value (Chi2) | Community enrolments | RHF enrolments | P value (Chi2) | Community enrolments | RHF enrolments | P value (Chi2) |
| | N = 1,939 | N = 2,269 | | N = 217 | N = 834 | | N = 225 | N = 2,499 | |
| | n (%) | n (%) | | n (%) | n (%) | | n (%) | n (%) | |
| **Administration of at least 1 dose of an inj. antimalarial[1]** | 1,613 (83.2) | 1,908 (84.1) | 0.430 | 208 (95.9) | 776 (93.1) | 0.132 | 209 (92.9) | 2,374 (95.0) | 0.171 |
| Artesunate | 1,379 (71.1) | 1,349 (59.5) | <0.001 | 202 (93.1) | 736 (88.3) | 0.040 | 201 (89.3) | 2,286 (91.5) | 0.275 |
| Artemether | 11 (0.6) | 34 (1.5) | 0.003 | 6 (2.8) | 43 (5.2) | 0.137 | 0 (0.0) | 1 (0.0) | 0.764 |
| Quinine | 229 (11.8) | 539 (23.8) | <0.001 | 4 (1.8) | 8 (1.0) | 0.275 | 10 (4.4) | 99 (4.0) | 0.723 |
| **In-hospital administration of at least 1 dose of each an inj. and an oral antimalarial[1]** | 1,300 (67.0) | 1,499 (66.1) | 0.502 | 0 (0.0) | 28 (3.4) | 0.006 | 74 (32.9) | 1,138 (45.5) | <0.001 |
| ACT[2] | 1,100 (56.7) | 1,017 (44.8) | <0.001 | 0 (0.0) | 28 (3.4) | 0.006 | 74 (32.9) | 1,137 (45.5) | <0.001 |
| Artemether-lumefantrine[2] | 91 (4.7) | 91 (4.0) | 0.278 | 0 (0.0) | 27 (3.2) | 0.007 | 74 (32.9) | 1,137 (45.5) | <0.001 |
| Artesunate-amodiaquine[2] | 1,010 (52.1) | 930 (41.0) | <0.001 | 0 (0.0) | 1 (0.1) | 0.610 | 0 (0.0) | 0 (0.0) | NA |
| Quinine | 205 (10.6) | 489 (21.6) | <0.001 | 0 (0.0) | 0 (0.0) | NA | 0 (0.0) | 1 (0.0) | 0.764 |
| **Administration of ACT only** | 90 (4.6) | 57 (2.5) | <0.001 | 0 (0.0) | 4 (0.5) | 0.307 | 6 (2.7) | 65 (2.6) | 0.953 |

Number and column % of children receiving injectable and oral antimalarial treatment, by country and enrolment location.

[1] More than 1 type of antimalarial may have been administered.

[2] Appropriate medication.

ACT, artemisinin-based combination therapy; DRC, Democratic Republic of the Congo; RHF, referral health facility.

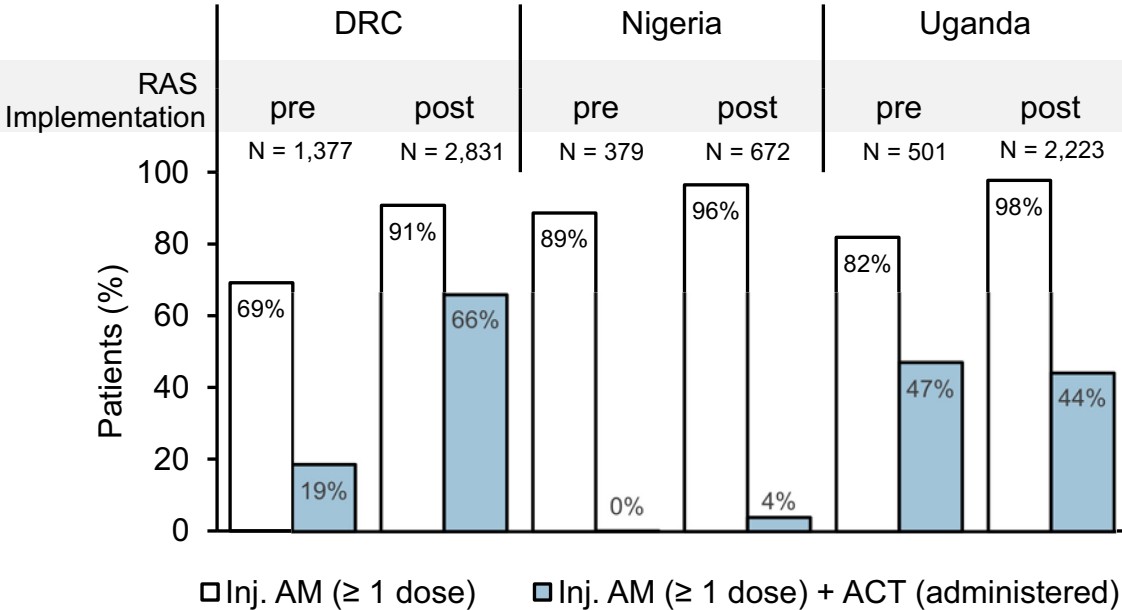

**Fig 2. Appropriateness of antimalarial medication provided to children diagnosed with severe malaria before and after the implementation of RAS, by country and by RAS implementation period (%).** Treatment of admitted children referred by a community-based provider or directly attending an RHF was assessed before and after the roll-out of RAS (pre vs. post). Proportion of children administered at least 1 dose of an injectable antimalarial (artesunate, artemether, or quinine; white bars) and at least 1 dose of an injectable antimalarial and an in-hospital ACT consisting of either ALU or ASAQ (blue bars). ACT, artemisinin-based combination therapy; ALU, artemether-lumefantrine; AM, antimalarial; ASAQ, artesunate-amodiaquine; DRC, Democratic Republic of the Congo; RAS, rectal artesunate.

**Table 3. Patient, provider, caregiver, and facility correlates with antimalarial medication appropriateness according to the WHO malaria treatment guidelines.**

| | DRC | | Nigeria | | Uganda | |
|---|---|---|---|---|---|---|
| | aOR (95% CI)[a] | P value | aOR (95% CI)[a] | P value | aOR (95% CI)[a] | P value |
| **Patient variables** | | | | | | |
| **Age (years)** | | | | | | |
| <1 | 1.45 (1.11–1.88) | 0.006 | 0.15 (0.01–1.68) | 0.12 | 1.01 (0.67–1.52) | 0.96 |
| 1–2 | 1.45 (1.19–1.77) | <0.001 | 0.72 (0.21–2.47) | 0.60 | 0.95 (0.72–1.27) | 0.74 |
| 3 - < 5 | Ref | | Ref | | Ref | |
| **Sex** | | | | | | |
| Male | Ref | | Ref | | Ref | |
| Female | 0.97 (0.84–1.12) | 0.65 | 0.41 (0.15–1.14) | 0.09 | 0.85 (0.69–1.05) | 0.13 |
| **Weight (kg)** | | | | | | |
| <8 | 0.98 (0.75–1.30) | 0.91 | 1.99 (0.46–8.68) | 0.36 | 1.36 (0.92–2.00) | 0.13 |
| 8–10 | 1.00 (0.83–1.21) | 0.98 | 1.30 (0.35–4.89) | 0.70 | 1.06 (0.82–1.39) | 0.65 |
| >10 | Ref | | Ref | | Ref | |
| **Administration of RAS** | | | | | | |
| No (post-implementation, community) | Ref | | Ref | | Ref | |
| Yes (post-implementation, community) | 2.13 (1.55–2.92) | <0.001 | 0.53 (0.01–9.02) | 0.95 | 0.37 (0.14–0.96) | 0.04 |
| NA (Pre-implementation, RHF) | 1.69 (1.10–2.60) | 0.02 | 2.79 (0.10–7.43) | 0.78 | 0.95 (0.32–2.79) | 0.93 |
| **Enrolment at community provider** | | | | | | |
| No | Ref | | Ref | | Ref | |
| Yes | 1.15 (0.85–1.55) | 0.36 | 1.36 (0.15–4.76) | 0.86 | 1.57 (0.73–3.37) | 0.25 |
| **mRDT/blood slide result** | | | | | | |
| Negative/not done | Ref | | Ref | | Ref | |
| Positive | 1.00 (0.79–1.26) | 0.97 | 0.44 (0.17–1.14) | 0.09 | 2.32 (0.58–9.27) | 0.23 |
| **Provider variable** | | | | | | |
| **Costs incurred for** | | | | | | |
| Drugs | 0.81 (0.67–0.97) | 0.02 | 1.01 (0.06–14.3) | 0.82 | 0.24 (0.18–0.33) | <0.001 |
| Hospitalisation | 1.21 (1.02–1.42) | 0.03 | 1.52 (0.12–9.42) | 0.79 | 3.90 (2.77–5.48) | <0.001 |
| **Caregiver variable** | | | | | | |
| **Age (Years)** | | | | | | |
| <30 | 1.17 (1.00–1.38) | 0.05 | 0.53 (0.21–1.37) | 0.19 | 1.20 (0.96–1.49) | 0.12 |
| ≥30 | Ref | | Ref | | Ref | |
| **Other contextual factors** | | | | | | |
| **RAS implementation period (pre- vs. post-implementation)** | 6.29 (4.88–8.10) | <0.001 | 5.14 (0.57–46.4) | 0.15 | 1.03 (0.76–1.41) | 0.83 |
| **Health zone/LGA/District◇** | | | | | | |
| Kenge DRC/Fufore NG/Kole UG | 1.18 (0.96–1.44) | 0.07 | Ref | | Ref | |
| Kingandu DRC/Mayo Belwa NG/Oyam UG | 0.68 (0.55–0.85) | <0.001 | 1.42 (0.42–16.4) | 0.78 | 6.71 (5.13–8.77) | <0.001 |
| Ipamu DRC/Song NG/Kwania UG | Ref | | 8.28 (1.04–65.9) | 0.05 | 1.84 (1.12–3.03) | 0.02 |
| **Seasonality (rainy season)** | 1.03 (0.87–1.21) | 0.73 | 3.74 (0.74–18.8) | 0.11 | 0.66 (0.52–0.86) | 0.002 |

[a] aOR, 95% CI and p value obtained from logistic models. Adjusted for covariates shown accounting for clustering at RHF level, participants with missing data for any of the variables were excluded.

◇Corresponds to Health zones in DRC (Kenge, Kingandu, Ipamu*)/LGA in Nigeria (Fufore*, Mayo Belwa, Song)/District in Uganda (Kole*, Oyam, Kwania) (* = Ref).

aOR, adjusted odds ratio; CI, confidence interval; DRC, Democratic Republic of the Congo; mRDT, rapid diagnostic test for malaria; RAS, rectal artesunate.

treatment if administered RAS (aOR = 0.37, 95% CI 0.14 to 0.96). The low numbers of community referrals and low levels of appropriate medication provision observed in Nigeria did not allow to compute estimates for these indicators.

In DRC, children admitted during the post-implementation period were much more likely to receive appropriate treatment compared to admissions during the pre-implementation phase (OR = 6.29, 95% CI 4.88 to 8.10). The odds for medication appropriateness were higher for children below 3 years (aOR = 1.45, 95% CI 1.19 to 1.77). No such differences were observed in the other 2 countries.

Other predictors for appropriate antimalarial medication included whether costs were incurred to caregivers: Both in Uganda and DRC, payable hospitalisation was positively correlated with receiving appropriate treatment, while the odds were lower if caregivers had to pay for medication. Again, the low number of events in Nigeria did not permit estimating these predictors.

A sensitivity analysis using multiple imputation methods for weight showed no differences in the results (S2 Table).

## In-hospital versus home ACT treatment

Study countries differed considerably in the way of providing follow-on ACT (Table 4). In DRC, ACT was usually started as an in-hospital therapy following completion of injectable treatment (78.7%, 1,314/1,669) and completed in 49.3% (822/1,669) of the cases while the patient was still hospitalised. By contrast, only 1.7% (7/421) in Nigeria and 45.7% (617/1,349) in Uganda received any ACTs as in-patients, as the usual process was to receive a prescription to buy an ACT at discharge: 54.4% (229/421) of admitted children in Nigeria and 53.0% (715/1,349) in Uganda.

## Antimalarial treatment compliance

Among the post-implementation subsample with detailed dosage information, the vast majority treated with an injectable antimalarial received at least 3 doses (3,220/3,337 (96.5%);

**Table 4. Provision of in-hospital vs. post-discharge ACT medication.**

| | DRC | | | Nigeria | | | Uganda | | |
|---|---|---|---|---|---|---|---|---|---|
| | Community enrolments | RHF enrolments | P value (Chi2) | Community enrolments | RHF enrolments | P value (Chi2) | Community enrolments | RHF enrolments | P value (Chi2) |
| | N = 846 | N = 823 | | N = 113 | N = 308 | | N = 132 | N = 1,217 | |
| | n (%) | n (%) | | n (%) | n (%) | | n (%) | n (%) | |
| **Provision of ACT treatment** | | | 0.021 | | | 0.200 | | | 0.002 |
| No ACT | 145 (17.1) | 170 (20.7) | | 54 (47.8) | 131 (42.5) | | 1 (0.8) | 15 (1.2) | |
| ACT treatment completed at facility* | 425 (50.2) | 397 (48.2) | | 0 (0.0) | 0 (0.0) | | 7 (5.3) | 165 (13.6) | |
| Started at facility, to be completed at home ◊ | 251 (29.7) | 241 (29.3) | | 0 (0.0) | 7 (2.3) | | 33 (25.0) | 412 (33.9) | |
| Received prescription to buy from pharmacy˚ | 5 (0.6) | 9 (1.1) | | 59 (52.2) | 170 (55.2) | | 91 (68.9) | 624 (51.3) | |
| Other | 20 (2.4) | 6 (0.7) | | 0 (0.0) | 0 (0.0) | | 0 (0.0) | 1 (0.1) | |
| missing | 4 (0.5) | 0 (0.0) | | 0 (0.0) | 0 (0.0) | | 0 (0.0) | 6 (0.5) | |

Number and % of distribution of modalities of receiving ACT treatment, by country and enrolment location; missing data rows are number and column %.

* Includes 3 children who received artesunate + mefloquine.

◊ Includes 7 children who received artesunate + mefloquine and 70 observations with missing specification of type of ACT given.

˚ Includes 1 child who received a prescription for dihydroartemisinin + piperaquine and 37 observations with missing specification of type of ACT prescribed.

ACT, artemisinin-based combination therapy; DRC, Democratic Republic of the Congo; RHF, referral health facility.

S3 Table). In DRC, 76.2% (1,274/1,673) of children were treated with both an injectable and an ACT during admission (Fig 3) and including post-discharge prescriptions did not much change this percentage (76.9% (1,287/1,673; Table 4). As noted above for Nigeria in the full dataset analysis, the level of compliance of in-hospital treatment administration was very low (1.2%, 5/421); antimalarial prescription compliance was elevated but remained low at only 45.6% (192/421). In Uganda, including ACT prescriptions in the estimate for treatment compliance more than doubled the percentage (44.7% (606/1,355) versus 97.5% (1,321/1,355), and compliance of in-hospital treatment administration to children referred from the community was lower as compared to direct RHF attendances (29.5% (39/132) versus 46.4% (567/1,223), $P < 0.001$; Fig 3). Since community referrals were more likely to receive a prescription, this difference vanished for prescription compliance (98.5% (130/132) versus 97.4% (1,191/1,223)).

## Discussion

Pre-referral RAS administered in the community or at the PHC level is intended to rapidly initiate effective malaria treatment in hard-to-reach locations before the patient is referred to a health facility with full case management capabilities. Adequate post-referral management is critical to ensure complete patient cure and avoid death and persisting disability.

In line with previous studies [7–9,19], the majority of children treated for severe malaria at an RHF received an injectable antimalarial, usually artesunate. By contrast, our results show that completing a full course of an ACT (a central component of the WHO recommendation) is highly unsatisfactory. Published results for this indicator vary greatly between 4.8% in Uganda [9] and 43.4% in Nigeria [20], though these reports do not specify whether ACTs were prescribed for in-hospital administration or at discharge. In our study, methods of ACT provision varied: ACTs were either directly administered at the RHF, or patients were discharged with a prescription for home treatment, or a variation thereof.

Our data suggest that in DRC and Uganda, a fair proportion of children start their ACT course while they are still admitted, likely being dispensed the remaining doses at discharge. This was rarely the case in Nigeria, where children either only received a prescription or no ACT at all. Failure to provide a full course of an ACT in the RHF and only giving a prescription raises 2 major concerns about the treatment's effectiveness. Firstly, an incomplete treatment together with a lower referral completion of community-enrolled children [15] increases significantly the risk of an unfavourable health outcome, including the risk of dying. This may have contributed to our finding that the beneficial effect of pre-referral RAS on survival [21,22] could not be replicated in our "real-world" study settings [17]. We also found a worrying level of sickness and mRDT positivity at day 28 among enrolled children; however, our study set-up did not permit distinguishing between new and recrudescent infections [17]. Secondly, incomplete treatment results in artemisinin monotherapy (RAS and injectable artesunate) and a raised risk of artemisinin resistance development. The selection of *P. falciparum* harbouring artemisinin K13 resistance mutations was found in the context of the CARAMAL project in Uganda (Awor and colleagues, manuscript in preparation).

If treatment is provided as prescription, the patient's adherence is crucial to ensure effective antimalarial treatment. Studies on patient and caregiver adherence to antimalarial treatment guidelines found large variations among different countries, ranging from <50% to 100% [23–28]. Adherence was found to be influenced by whether ACTs were delivered by the public or the private sector, as well as by caregiver income [27]. It seems likely that adherence to ACT is higher if the drug is dispensed rather than provided as a prescription that needs to be filled by the caregiver.

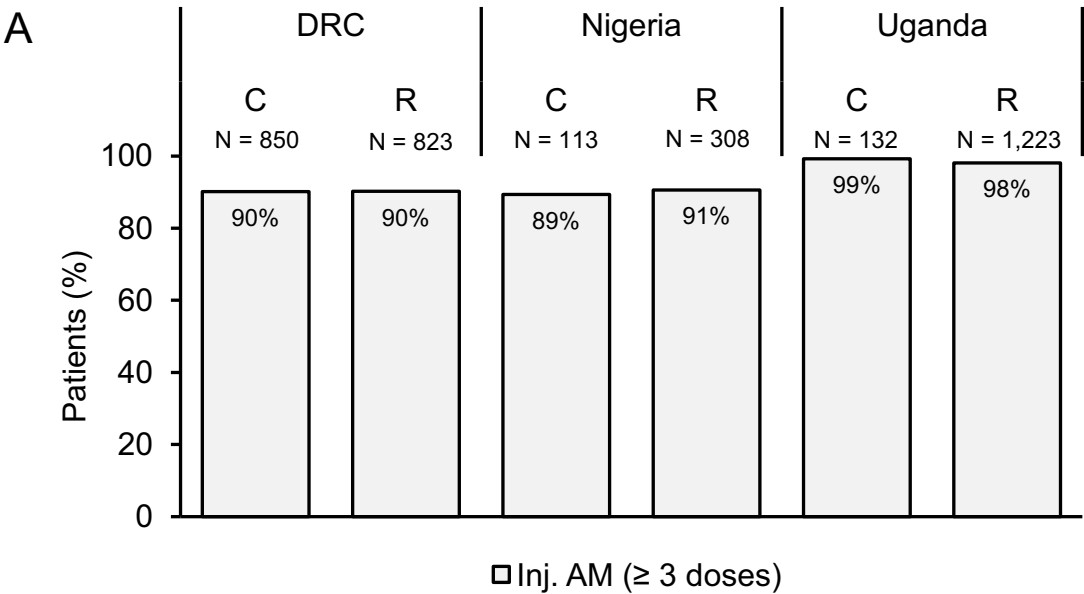

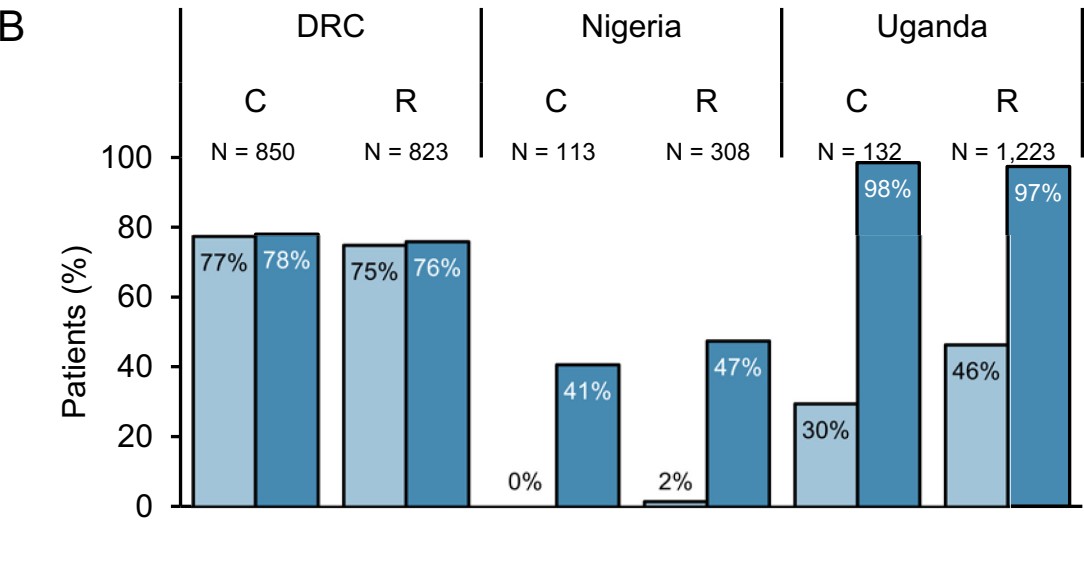

**Fig 3. Antimalarial treatment compliance for children diagnosed with severe malaria after the implementation of RAS, by country and by enrolling provider (%).** Antimalarial treatment administered in-hospital or prescribed at discharge to children referred by a CHW or PHC (C) or directly attending an RHF (R). (A) Administration of at least 3 doses of an injectable antimalarial (artesunate, artemether, or quinine). (B) Administration of at least 3 doses of an injectable antimalarial and in-hospital follow-on ACT (light blue bars) or in-hospital administered/at discharge prescribed/dispensed follow-on ACT (dark blue bars). Data collection period: Uganda and DRC: April 2019–July 2020, Nigeria: May 2019–July 2020. ACT, artemisinin-based combination therapy; AM, antimalarial; DRC, Democratic Republic of the Congo; RAS, Rectal Artesunate; RHF, Referral Health Facility.

Reason for discharging a child before the start of ACT therapy could include a limited bed capacity at RHFs, especially during rainy seasons or disease outbreaks, though a higher

treatment compliance during the rainy season seen in Nigeria suggests otherwise (Table 3). Coronavirus Disease 2019 restrictions in Nigeria and Uganda impacting people's movement and the supply chain between March and July 2020 likely affected the running of these facilities. ACT stock-outs at RHFs may also have led to an increase of prescriptions though our annual cross-sectional healthcare provider surveys did not reveal major stock-outs. Further, socioeconomic factors may have influenced differently hospitalisation duration and treatment patterns among community and RHF enrolments. Such factors may account for our finding that referral cases in Uganda were more likely to receive a prescription rather than in-hospital ACT treatment compared to children directly attending an RHF. Early hospital discharge also happens if inpatient medical care is no longer required (e.g., the child is able to swallow and further treatment may be continued at home). Discharging therefore relieves the burden on both the RHF (beds) and the family (hospitalisation costs).

The relatively high rate of treatment compliance observed in DRC during the post-implementation period may be a result of a number of supportive interventions implemented to facilitate the roll-out and uptake of RAS [14,18]. In particular, this included distributing injectable artesunate to RHFs to limit use of the inferior quinine [29–31].

Administration of pre-referral RAS did have a positive impact on whether a child received appropriate treatment in DRC. This and increased use of injectable artesunate in referrals compared to RHF enrolments suggests an increased awareness and may be due to targeted health worker training in the frame of RAS roll-out [14]. In Uganda, RAS use was negatively correlated with appropriateness of antimalarial treatment. This finding raises important concerns, namely that the full course of antimalarial treatment for severe disease may no longer be considered necessary by healthcare workers after a single dose of RAS and a rapid improvement of the child's episode. Our finding that the likelihood of receiving appropriate treatment increased if caregivers had to pay a hospitalisation fee could imply the provision of better quality services if payment is made by patients.

This study was conducted in 3 countries with high malaria burden but very different health systems contexts which may, at least in part, explain the differences in treatment compliance observed. All 3 countries base their severe malaria treatment guidelines on the WHO recommendations. To comply with these guidelines, we identify the following prerequisites for provision of compliant treatment for severe malaria. First, as evidenced by our DRC data, availability of the recommended drugs, in particular artesunate and ACTs, must be guaranteed. Second, health workers must know and comply with the treatment guidelines; any incentives nor non-compliant treatment (e.g., financial) should be counteracted. Sensitisation for the need to complete treatment even if symptoms have improved is of paramount importance. Third, if the ACT course cannot be completed at the RHF, drug dispensation seems more favourable than a prescription. And finally, mechanisms to monitor adherence of home treatment should be implemented to ensure adequate treatment completion (e.g., follow-up home visit by CHW).

While our study lacks detailed information on patient adherence with prescribed ACTs, the prospective recording of case management during admission provided more accurate information than other studies using a retrospective design. The present multicountry study allowed us to include an unprecedented large sample of severely ill children from very distinct contexts (in terms of disease burden, health system, access to healthcare, etc.) while investigating the effect of introducing pre-referral RAS.

At the same time, the surprisingly different contexts in each country led to different results for key parameters and also implied slight differences in the detail of data collected. This impacted negatively the depth in which certain findings could be analysed across countries. Information on caregiver socioeconomic parameters like education, employment status,

income were not collected and could not be included as predictors in the regression model. Training of study staff and standardised record forms were implemented to minimise observer bias and differences between countries. We cannot rule out the possibility of a Hawthorne effect due to the study staff's presence, potentially leading to an overestimation of the quality of care.

Finally, this study was limited to evaluating the treatment of severe malaria based on the local clinicians' diagnosis of "severe malaria." This diagnosis was not independently confirmed because it was neither possible nor desirable (to minimise the Hawthorne effect mentioned above) to place a fully qualified clinician in each RHF.

## Conclusion

Pre-referral RAS for children in hard-to-reach locations can only be an effective addition to a functioning continuum of care for severe malaria, if post-referral treatment is adequate [32]. While parenteral treatment was generally administered correctly and reliably, we found that the provision of ACTs to complete treatment was often not followed or left to the discretion of the caregiver for home treatment. This resulted in a low overall treatment compliance, entailing a danger of poor treatment outcomes and an increased risk of resistance development. In order to effectively integrate pre-referral RAS into the continuum of care for severe malaria, health system deficiencies need to be addressed and health worker compliance strengthened to ensure the provision of effective, life-saving post-referral treatment.

## Supporting information

**S1 File. CARAMAL protocol.**
(DOCX)

**S2 File. STROBE checklist.**
(DOCX)

**S3 File. PLOS policy and questionnaire.**
(DOCX)

**S1 Table. Summary characteristics of surveyed patients and exposure variables (subsample, post-implementation only).**
(DOCX)

**S2 Table. Patient, provider, caregiver, and facility correlates with antimalarial medication appropriateness according to the WHO malaria treatment guidelines (sensitivity analysis).**
(DOCX)

**S3 Table. Antimalarial treatment administration and prescription compliance: number of doses of injectable antimalarials administered and follow-on ACT administration/prescription (subsample, post-implementation only).**
(DOCX)

## Acknowledgments

The study team would like to sincerely thank all the children and their caregivers who agreed to participate in this study; the health workers and local and national health authorities who provided their support; our study teams of the Kinshasa School of Public Health (DRC), Akena Associates Ltd. (Nigeria), and the Makerere University School of Public Health (Uganda); and the colleagues of the local CHAI and UNICEF offices.

## Author Contributions

**Conceptualization:** Aita Signorell, Manuel W. Hetzel, Valentina Buj, Christian Lengeler, Christian Burri.

**Data curation:** Aita Signorell, Proscovia Athieno, Joseph Kimera, Gloria Tumukunde, Irene Angiro, Babatunde K. Akano, Kazeem Ayodeji, Charles Okon, Giulia Delvento, Tristan T. Lee, Nina C. Brunner.

**Formal analysis:** Aita Signorell, James Okuma.

**Investigation:** Phyllis Awor, Jean Okitawutshu, Antoinette Tshefu, Elizabeth Omoluabi, Proscovia Athieno, Joseph Kimera, Gloria Tumukunde, Irene Angiro, Jean-Claude Kalenga, Babatunde K. Akano, Kazeem Ayodeji, Charles Okon, Ocheche Yusuf.

**Methodology:** Aita Signorell, Phyllis Awor, Jean Okitawutshu, Antoinette Tshefu, Elizabeth Omoluabi, Manuel W. Hetzel, Nina C. Brunner, Christian Lengeler, Christian Burri.

**Project administration:** Aita Signorell, Phyllis Awor, Jean Okitawutshu, Antoinette Tshefu, Elizabeth Omoluabi, Manuel W. Hetzel, Proscovia Athieno, Joseph Kimera, Gloria Tumukunde, Irene Angiro, Ocheche Yusuf, Nadja Cereghetti, Christian Lengeler, Christian Burri.

**Supervision:** Aita Signorell, Phyllis Awor, Jean Okitawutshu, Antoinette Tshefu, Elizabeth Omoluabi, Manuel W. Hetzel, Proscovia Athieno, Joseph Kimera, Gloria Tumukunde, Irene Angiro, Jean-Claude Kalenga, Babatunde K. Akano, Kazeem Ayodeji, Charles Okon, Ocheche Yusuf, Nina C. Brunner, Nadja Cereghetti, Christian Lengeler, Christian Burri.

**Validation:** Aita Signorell.

**Visualization:** Aita Signorell.

**Writing – original draft:** Aita Signorell.

**Writing – review & editing:** Aita Signorell, Phyllis Awor, Jean Okitawutshu, Antoinette Tshefu, Elizabeth Omoluabi, Manuel W. Hetzel, Proscovia Athieno, Joseph Kimera, Gloria Tumukunde, Irene Angiro, Jean-Claude Kalenga, Babatunde K. Akano, Kazeem Ayodeji, Charles Okon, Ocheche Yusuf, Giulia Delvento, Nina C. Brunner, Mark J. Lambiris, James Okuma, Nadja Cereghetti, Valentina Buj, Theodoor Visser, Harriet G. Napier, Christian Lengeler, Christian Burri.

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
