## [Editor Report · Decision Letter 0]

28 Jul 2022

Dear Dr Signorell, 

Thank you for submitting your manuscript entitled "Health worker compliance with severe malaria treatment guidelines in the context of implementing pre-referral rectal artesunate: an operational study in three high burden countries Quality of post-referral severe malaria treatment" for consideration by PLOS Medicine.

Your manuscript has now been evaluated by the PLOS Medicine editorial staff and I am writing to let you know that we would like to send your submission out for external peer review.

Please re-submit your manuscript within two working days, i.e. by Aug 01 2022 11:59PM.

Kind regards,

Beryne Odeny, PhD

PLOS Medicine

---

## [Decision Letter · Decision Letter 1]

29 Sep 2022

Dear Dr. Signorell,

Thank you very much for submitting your manuscript "Health worker compliance with severe malaria treatment guidelines in the context of implementing pre-referral rectal artesunate: an operational study in three high burden countries" (PMEDICINE-D-22-02521R1) for consideration at PLOS Medicine. 

[LINK]

In light of these reviews, I am afraid that we will not be able to accept the manuscript for publication in the journal in its current form, but we would like to consider a revised version that addresses the reviewers' and editors' comments. Obviously we cannot make any decision about publication until we have seen the revised manuscript and your response, and we plan to seek re-review by one or more of the reviewers. 

We expect to receive your revised manuscript by Oct 20 2022 11:59PM. Please email us (plosmedicine@plos.org) if you have any questions or concerns.

We look forward to receiving your revised manuscript. 

Sincerely,

Beryne Odeny, 

PLOS Medicine

plosmedicine.org

1) We note that you conducted research or obtained samples in a foreign country. Did you consider including a local author as first or last author? If not, we recommend that you consider doing so in line with ICMJE's authorship requirements (https://www.icmje.org/recommendations/browse/roles-and-responsibilities/defining-the-role-of-authors-and-contributors.html). PLOS also has a parachute research policy which aims to promote collaboration and inclusivity in global health research. You are required to complete PLOS’ questionnaire on inclusivity in global research and submit it with your revised paper. The policy and questionnaire can be found at https://journals.plos.org/plosone/s/best-practices-in-research-reporting.

2) Please revise your title according to PLOS Medicine's style. Your title must be nondeclarative and not a question. It should begin with main concept if possible. Please place the study design only in the subtitle (i.e., after a colon) and sample description before the colon. For example, ““Health worker compliance with severe malaria treatment guidelines in the context of implementing pre-referral rectal artesunate in the Democratic Republic of Congo, Nigeria and Uganda: an operational study” 

3) Please include specific countries in the title instead of “three high burden countries”

4) Abstract:

a) Background: The final sentence should clearly state the study question.

b) In the Methods and Findings, please include the study design, the target population, years during which the study took place, length of follow up, and main outcome measures.

c) Line 39, should read, “It included…”

d) Please include the important dependent variables that are adjusted for in the analyses.

e) Please include p values, as appropriate, in addition to 95% CIs.

f) Please ensure that all numbers presented in the abstract are present and identical to numbers presented in the main manuscript text.

g) In the last sentence of the Abstract Methods and Findings section, please describe the main limitation(s) of the study's methodology.

6) Please include the relevant protocol or prospectively written document with your revised manuscript as a Supporting Information file to be published alongside your study and cite it in the Methods section. A legend for this file should be included at the end of your manuscript. 

7) Please make sure that the Methods section transparently describes when analyses were planned, and when/why any data-driven changes to analyses took place, including those made in response to peer review comments-- should be identified as such in the Methods section of the paper, with rationale.

8) As requested by reviewers, please provide further context and detail regarding the accompanying CARAMAL project.

9) Please ensure that the study is reported according to the STROBE and include the completed STROBE checklist as Supporting Information. Please add the following statement, or similar, to the Methods: "This study is reported as per the Strengthening the Reporting of Observational Studies in Epidemiology (STROBE) guideline (S1 Checklist)."

10) Did you consider adjustment of maternal factions like education, employment status, age – factors which may affect completion of treatment. If not, please highlight this as a limitation?

11) Please include line numbers after in the results and discussion 

12) Table 1: under the subheading “Health Zone/ LGA/ District,” please explain what the names mean and which country representation they stand for, e.g., “Kenge….”

13) Instead of pre- and post-RAS, please refer to “pre- and post- intervention” period to avoid confusion 

14) Please consistently provide both 95% CIs and p values in results and tables.

15) Please state p <0.001, not p <0.000

16) Please define all abbreviations in the tables e.g., mRDT, RAS

17) Figure 2 & 3: please explain the meaning of the bars and asterisk in the footnotes

18) Please replace the term “compliance” with “adherence” where it is used to refer to treatment adherence.

19) Given the limitations of the observational study design, in the discussion and conclusions, please remove 

causal language such as “impact,” “effect,” and “leading to poor treatment outcomes ….” Refer to associations instead.

20) Discussion: paragraph 7, line 7, should read “… single dose of RAS…”

21) Conclusion: Please provide specific implications for policy and clinical practice as substantiated by the results.

22) Please remove the ‘Funding,” and “Conflict of interest” statements from the end of the main text. In the event of publication, this information will be published as metadata based on your responses to the submission form.

23) References:

a) Please select the PLOS Medicine reference style in your citation manager. In-text reference call outs should be presented as follows noting the absence of spaces within the square brackets, e.g., "... services [1,2]."

b) Please ensure that journal name abbreviations consistently match those found in the National Center for Biotechnology Information (NCBI) databases. https://journals.plos.org/plosmedicine/s/submission-guidelines#loc-references. e.g. PLoS Med.

c) Please ensure all weblinks are accessible and have access dates.

Comments from the reviewers:

Reviewer #1: The paper describes one aspect of the CARAMAL project which has made a major negative impact on WHO policy. The paper is critical to understand why the project has provided counterintutitive results. The publication of the paper seems to me an urgent priority.

This paper has to deal with two major challenges:

We are only getting one slice of the Salami. The treatment compliance described in the paper under review is critical and relevant for the outcome of the project yet without knowing the project the reader may be baffled regarding its relevance. It is my understanding that the key paper by Hetzel et al is about to be published. To help the reader grasp the relevance of the paper the authors should summarize the key findings of the project i.e. increased mortality after RAS roll-out in the introduction and/or in a box regarding 'research in context'. 

The second challenge are the major discrepancies in research findings between countries. If findings and (as the authors inform us) some of the methods between countries are so different it makes little or no sense at all to pool the findings. For the reader it would be more convenient being able just to look at one set of pooled summary data but that would provide a wrong impression. It is quite possible that not all readers of the journal are familiar with these statistical facts hence it may be worthwhile to explain why the country findings are reported separately? More importantly the authors can't resist to report repeatedly pooled data from the three study countries. It will be interesting to see whether the statistical reviewer considers this legitimate.

Please state what kind of a study this is e.g. prospective, observational?

Abstract: the conclusions are based on data from the project but not from data included in the paper. if the authors want to include these conclusions and I think they should, they also have to state the mortality findings of the caramal project in their abstract.

Methods "Information on the use of pre-referral RAS was consolidated from different data sources through the different points of contact with the healthcare system (CHW, PHC, RHF) and from a caregiver's interview on day 28." please explain how this was done, i.e. data management? How were discrepancies dealt with? 

Definitions: pls. explain what you mean by 'three dimensions'?

Statistics: "Costs incurred to caregivers for hospitalisation or medication were analysed as provider-level predictors" the paper under review does not include data on costs.

Results: there is a major drop out of participants from 14,911 provisionally enrolled, to 7,983 included in the analysis, and 3,449 with an 'update data collection form'. I find it hard to understand how much data are missing? Which data are in the updated form which are not in the other form and what are the implications for the generalizability of the data?

The authors sometimes refer to compliance with treatment, a term I am familiar with and then they use 'compliant treatment'. I may not be the only one who is confused by 'compliant treatment'. It sounds wrong to me. I apologize if that is due to my ignorance.

Discussion: the word 'obviously' in the sentence "Adequate post-referral management is obviously critical to ensure complete patient cure and avoid death and persisting disability." strikes me as downright cruel. I can't help wondering if this is so obvious why didn't the project assure compliance with the post-referral management? The flippant way the sentence is worded the authors may open an ethics quagmire very difficult to get out of. 

The conclusions read well but are not based on the data presented in the paper. It is essential that the authors include these data in one form or another in this paper to allow the reader to understand what they mean.

Tables: the authors have moved all tables to the supplementary section. Why not include the four tables in the text?

Figures should use colour.

Figure 2: pls. revise the title of the y-axis. Patients treatment (%) is not informative.

Figure 3B: pls. revise the legend to clearly indicate the difference between the two groups.

Reviewer #2: See attachment

Michael Dewey

Reviewer #3: Adequate and complete follow-up treatment at referral facilities is a critical factor in the context of implementation of pre- referral rectal artesunate. This requires strong health systems, including well-functioning referral facilities, both in terms of availability of trained health workers and commodities for the treatment of severe malaria. This paper presents findings on health workers' adherence to severe malaria treatment guidelines in referral facilities in the context of implementing pre-referral rectal artesunate. 

The paper is well writtend. Below are specific comments.

1. Although the authors refer to the main paper for more detailed information on the methods for this manuscript, the components of the methods relevant to this substudy should be clearly described. How were RAS children tracked up to the study RHFs? What data collection tools were used at different time points: pre and post RAS? Exactly what supportive measures were in place for the deployment of RAS at the different sites to better understand the different contexts, etc.

2. Page 5-6, line 105-106: "Patients provisionally enrolled at the community level were excluded from this analysis if the diagnosis of severe malaria was not confirmed in the RHF." The authors described the signs and symptoms of severe malaria in the inclusion criteria for community-level workers. But for severe patients in the RHF, how was severe malaria diagnosed? Were laboratory tests (RDT or microscopy) performed to confirm malaria? What was the national guideline for diagnosing severe malaria that RHF staff in different countries should follow? These need to be described for each country based on their treatment guidelines. 

3. Page 6, lines 125-127: "General treatment compliance was computed for the entire study population, while a refined treatment assessment was done in a sub-population for which the updated data collection form was used." The terms "General treatment compliance" and "refined treatment assessment" need to be defined. Authors refer here also "updated data collection form" here. This implies that different data collection forms were used at different time points in the study. These need to be described in detail. 

4. Page 10, lines 173-174: "Across the full study period, most of the children were treated with an injectable antimalarial (DRC 83.7%, Nigeria 93.6% and Uganda 94.8%; Supplementary Table 2)." To clarify the statement, change to "Across the full study period, most of the children were treated with an injectable antimalarial at RHFs (DRC 83.7%, Nigeria 93.6% and Uganda 94.8%; Supplementary Table 2)." 

5. Page 10, lines 173-179: In the statements "Across the full study period, most of the children were treated with an injectable antimalarial (DRC 83.7%, Nigeria 93.6% and Uganda 94.8%; Supplementary Table 2). In 86.8% of these cases, injectable artesunate was administered. During the post- RAS period, administration of parenteral antimalarials was higher (94.1%, artesunate 87.8%) than in the pre- RAS period (75.2%, artesunate 50.0%; Fig 2, Supplementary Table 2). In DRC, the use of intravenous quinine was still common (18.3% among all children) though it was gradually replaced by artesunate throughout study duration (34.6% pre- RAS, 10.3% post- RAS)." These data refer to all admitted severe patients in RHFs. It would be informative to provide data on parenteral treatment of RAS treated patients from the community. 

6. Page 10, lines 194-196: "Throughout the full study period, 42.0% of the children received both an injectable and an ACT, i.e. compliant treatment, during admission (Supplementary Table 2). While only 2.7% of the children in Nigeria received oral follow-on treatment, 44.5% did so in Uganda and 50.3% in DRC." The term "compliant treatment" has a sligh .

7. RAS is a relatively new intervention and requires a context-specific new community-level delivery approach. The authors report that the use of pre-referral RAS had a positive impact on whether a child received compliant treatment in DRC and a negative impact in Uganda. They linked this to supportive interventions in the DRC, including training of health workers at RHF, provision of artesunate injectables, etc. This underscores the need to not only provide sustained support to community-based providers, but also to strengthen referral facilities. The authors did not elaborate on what we can learn from the three contexts studied and what they might suggest to improve appropriate follow-up treatment following RAS treatment.

[LINK]

---

## [Decision Letter · Decision Letter 2]

10 Jan 2023

Dear Dr. Signorell,

Thank you very much for re-submitting your manuscript "Health worker compliance with severe malaria treatment guidelines in the context of implementing pre-referral rectal artesunate in the Democratic Republic of the Congo, Nigeria and Uganda: an operational study" (PMEDICINE-D-22-02521R2) for review by PLOS Medicine.

I have discussed the paper with my colleagues and the academic editor and it was also seen again by three reviewers. I am pleased to say that provided the remaining editorial and production issues are dealt with we are planning to accept the paper for publication in the journal.

[LINK]

We look forward to receiving the revised manuscript by Jan 17 2023 11:59PM.   

Sincerely,

Callam Davidson, 

Senior Editor 

PLOS Medicine

plosmedicine.org

Requests from Editors:

Author Summary:

* Please shorten your Author Summary such that is contains 2-3 single sentence bullet points per question. Questions 1 and 2 can be trimmed to include only essential details and bullet points can be combined where appropriate. 

* Please include headline numbers in the summary, including sample size and key findings. 

* Please place your Author Summary in the main manuscript, after the Abstract and before the Introduction.

Has the CARAMAL protocol been published as part of previous work? If not, we would request that the protocol is included in the Supporting Information, in the interest of full transparency (unless there are any legal or ethical concerns associated with this inclusion). If already published (e.g., as Supporting Information for a prior paper), please feel free to cite this publication instead.

Abstract Methods and Findings: Please include the important dependent variables that are adjusted for in the analyses.

Comments from Reviewers:

Reviewer #1: thank you for taking the time to address my suggestions.

Reviewer #2: The authors have addressed my points but I have one remaining minor point. I think the multiple imputation performed as a sensitivity analysis should be shown and not reported as "not shown" it could go into the supplement obviously.

Michael Dewey

Reviewer #3: The authors addressed all the queries satisfactorily.

[LINK]

---

## [Editor Report · Decision Letter 3]

27 Jan 2023

Dear Dr. Signorell,

Thank you very much for re-submitting your manuscript "Health worker compliance with severe malaria treatment guidelines in the context of implementing pre-referral rectal artesunate in the Democratic Republic of the Congo, Nigeria and Uganda: an operational study" (PMEDICINE-D-22-02521R3) for review by PLOS Medicine.

The remaining issues that need to be addressed are listed at the end of this email. Please take these into account before resubmitting your manuscript. In revising the manuscript for further consideration here, please ensure you address the specific points made by each reviewer and the editors. In your rebuttal letter you should indicate your response to the reviewers' and editors' comments and the changes you have made in the manuscript. Please submit a clean version of the paper as the main article file. A version with changes marked must also be uploaded as a marked up manuscript file.

We hope to receive your revised manuscript within 3 working days. Please email us (plosmedicine@plos.org) if you have any questions or concerns.

We look forward to receiving the revised manuscript by Feb 08 2023 11:59PM.   

Sincerely,

Callam Davidson, 

Associate Editor 

PLOS Medicine

plosmedicine.org

Requests from Editors:

As previously requested by the statistical reviewer, please include the multiple imputation performed as a sensitivity analysis in the Supporting Information and cite the item in your Results.

Thank you for providing your protocol. In the interest of transparency, it is journal policy to request that authors submit prospective protocols, where they exist, to be published alongside the manuscript once accepted. Please confirm that the Sponsor has approved publication of the protocol, and that you have redacted any sensitive information (i.e., personal contact details of study personnel).

---

## [Editor Report · Decision Letter 4]

2 Feb 2023

Dear Dr Signorell, 

On behalf of my colleagues and the Academic Editor, Professor Lorenz Von Seidlein, I am pleased to inform you that we have agreed to publish your manuscript "Health worker compliance with severe malaria treatment guidelines in the context of implementing pre-referral rectal artesunate in the Democratic Republic of the Congo, Nigeria and Uganda: an operational study" (PMEDICINE-D-22-02521R4) in PLOS Medicine.

When completing the formatting changes, please also address the following editorial comments:

* Please ensure you define abbreviations in your Author Summary on first use.

* Please confirm that URL in your Data Availability Statement will become active prior to publication.

PRESS

Sincerely, 

Callam Davidson 

Associate Editor 

PLOS Medicine